# Trio-based whole exome sequencing in patients with suspected sporadic inborn errors of immunity: A retrospective cohort study

Anne Hebert[1], Annet Simons[1], Janneke HM Schuurs-Hoeijmakers[1], Hans JPM Koenen[2], Evelien Zonneveld-Huijssoon[3], Stefanie SV Henriet[4], Ellen JH Schatorjé[5], Esther PAH Hoppenreijs[5], Erika KSM Leenders[1], Etienne JM Janssen[6], Gijs WE Santen[7], Sonja A de Munnik[1], Simon V van Reijmersdal[1], Esther van Rijssen[2], Simone Kersten[1], Mihai G Netea[8,9], Ruben L Smeets[2,10], Frank L van de Veerdonk[8], Alexander Hoischen[1,8]*, Caspar I van der Made[1,8]

[1]Department of Human Genetics, Radboud Institute of Molecular Life Sciences (RIMLS), Radboud University Medical Center, Nijmegen, Netherlands; [2]Department of Laboratory Medicine, Laboratory for Medical Immunology, Radboud University Medical Center, Nijmegen, Netherlands; [3]Department of Genetics, University of Groningen, University Medical Center Groningen, Groningen, Netherlands; [4]Department of Pediatric Infectious Diseases and Immunology, Amalia Children's Hospital, Radboud Center for Infectious Diseases (RCI), Radboud University Medical Center, Nijmegen, Netherlands; [5]Department of Pediatric Rheumatology and Immunology, Amalia Children's Hospital, Radboud University Medical Center, Nijmegen, Netherlands; [6]Department of Clinical Genetics, Maastricht University Medical Center+, Maastricht, Netherlands; [7]Center for Human and Clinical Genetics, Leiden University Medical Center, Leiden, Netherlands; [8]Department of Internal Medicine and Radboud Center for Infectious Diseases (RCI), Radboud University Medical Center, Nijmegen, Netherlands; [9]Department for Immunology and Metabolism, Life and Medical Sciences Institute (LIMES), University of Bonn, Bonn, Germany; [10]Department of Laboratory Medicine, Laboratory for Diagnostics, Radboud University Medical Center, Nijmegen, Netherlands

*For correspondence: alexander.hoischen@radboudumc.nl

## Abstract

**Background:** De novo variants (DNVs) are currently not routinely evaluated as part of diagnostic whole exome sequencing (WES) analysis in patients with suspected inborn errors of immunity (IEI).

**Methods:** This study explored the potential added value of systematic assessment of DNVs in a retrospective cohort of 123 patients with a suspected sporadic IEI that underwent patient-parent trio-based WES.

**Results:** A (likely) molecular diagnosis for (part) of the immunological phenotype was achieved in 12 patients with the diagnostic in silico IEI WES gene panel. Systematic evaluation of rare, non-synonymous DNVs in coding or splice site regions led to the identification of 14 candidate DNVs in genes with an annotated immune function. DNVs were found in IEI genes (NLRP3 and RELA) and in potentially novel candidate genes, including PSMB10, DDX1, KMT2C, and FBXW11. The FBXW11 canonical splice site DNV was shown to lead to defective RNA splicing, increased NF-κB p65

signalling, and elevated IL-1β production in primary immune cells extracted from the patient with autoinflammatory disease.

**Conclusions:** Our findings in this retrospective cohort study advocate the implementation of trio-based sequencing in routine diagnostics of patients with sporadic IEI. Furthermore, we provide functional evidence supporting a causal role for *FBXW11* loss-of-function mutations in autoinflammatory disease.

**Funding:** This research was supported by grants from the European Union, ZonMW and the Radboud Institute for Molecular Life Sciences.

## Editor's evaluation

This is an important paper that reports on the diagnostic utility of TRIO-based whole-exome sequencing (WES) for patients with suspected monogenic inborn errors of immunity, which is supported by solid data. This manuscript will be of particular interest to medical geneticists, immunologists, and physicians working with patients with primary immunodeficiencies.

## Introduction

Although we inherit the vast majority of genomic variants from our parents, a small fraction of variants arises *de novo* during parental gametogenesis or after zygosis (*Acuna-Hidalgo et al., 2016*). The biological rate at which these variants develop in humans translates to an average of 50-100 *de novo* single nucleotide variants (SNVs) per genome per generation, only one or two of which affect coding regions (*Acuna-Hidalgo et al., 2016*; *Lynch, 2010*). *De novo* variants (DNVs) are often very rare or unique (absent from population databases) and have a higher *a priori* chance to be pathogenic than inherited variants (*Meyts et al., 2016*; *Veltman and Brunner, 2012*). In contrast to inherited variants, DNVs emerge between two generations and are subjected to minimal evolutionary selection pressure that would normally purify damaging mutations (*Acuna-Hidalgo et al., 2016*). DNVs affecting nucleotides or genes that have been targeted by strong purifying selection can therefore be highly damaging to their respective non-redundant biological functions, as has for example been shown for genes involved in innate immunity, an ancient host defence mechanism that developed under constant environmental selection pressure by microorganisms (*Veltman and Brunner, 2012*; *Quintana-Murci and Clark, 2013*).

Therefore, DNVs are important candidates to pursue as a cause for disease, particularly in rare, sporadic phenotypes (*Lynch, 2010*; *Veltman and Brunner, 2012*; *Vissers et al., 2010*). The presence of such candidate DNVs can be assessed by trio-based sequencing, in which the patient is sequenced together with the (healthy) parents (*Acuna-Hidalgo et al., 2016*). Most experience with the systematic diagnostic assessment of DNVs has been gained in the field of developmental disorders, in which DNVs have been shown to constitute up to 50% of disease-causing mutations (*Vissers et al., 2010*; *Martin et al., 2018*; *Kaplanis et al., 2020*). However, the contribution of DNVs in the pathogenesis of other disorders such as inborn errors of immunity (IEI) is less clear.

DNVs as the underlying cause in IEI patients have been widely reported in literature, but most of these mutations were determined to be *de novo* through subsequent segregation analysis and not by trio-based sequencing (*Stray-Pedersen et al., 2017*; *Arts et al., 2019*; *Rudilla et al., 2019*; *Bradshaw et al., 2018*; *Liu et al., 2011*). IEI can present at different stages of life with a variable phenotype ranging from recurrent, life-threatening infections to immune dysregulation and cancer (*Arts et al., 2019*; *Bousfiha et al., 2020*). Particularly in IEI patients with early-onset and severe complex phenotypes, there is an increased chance for an underlying causative DNV (*Veltman and Brunner, 2012*; *Vorsteveld et al., 2021*). Moreover, DNVs that arise post-zygotically or somatically are recognized as an important underlying cause for IEI patients with autoinflammatory disease (*Labrousse et al., 2018*; *de Inocencio et al., 2015*; *Mensa-Vilaro et al., 2016*; *Kawasaki et al., 2017*; *Holzelova et al., 2004*; *Aluri et al., 2021*; *Beck et al., 2020*; *van der Made et al., 2022*; *Zhou et al., 2012*). The potentially added value of systematic DNV assessment in IEI patients is supported by the findings of an international cohort study, which reported a diagnosis in 44% of cases after patient-parent trio sequencing, compared to 36% by single whole exome sequencing (WES) (*Stray-Pedersen et al., 2017*). However,

trio-based sequencing has not yet been implemented as part of the routine diagnostic procedure of IEI patients.

The current study has aimed to explore the potential added value of systematic assessment of DNVs in a retrospective cohort of 123 patients with a suspected, sporadic IEI that underwent trio-based WES.

## Materials and methods

### Patients and samples

We retrospectively screened patient-parent trios that were submitted to Genome Diagnostics at the Department of Human Genetics in the Radboud University Medical Center (RUMC) between May 2013 and November 2021. Patient-parent trios were selected for systematic DNV analysis when fulfilling the following inclusion criteria: (1) the patient's phenotype was sporadic, (2) the clinical description was suspect for an inborn error of immunity (IEI), and (3) the *in silico* IEI whole exome sequencing (WES) panel was requested and analysed. The *in silico* IEI gene panel of the RUMC is periodically updated after literature review and currently encompasses 456 genes (version DG3.1.0 *Radboudumc, 2021*). During the study period, the *in silico* IEI WES panel was analysed in 146 patient-parent trios, of which 123 trios met the inclusion criteria for our retrospective cohort study (*Figure 1*).

As described previously (*Arts et al., 2019*), patients and their parents provided written informed consent for *in silico* IEI WES gene panel analysis with or without exome-wide variant analysis that is in line with the diagnostic clinical question, as approved by the Medical Ethics Review Committee Arnhem-Nijmegen (2011/188 and 2020–7142). This research is in compliance with the principles of the Declaration of Helsinki (*World Medical, 2013*).

For the systematic DNV analysis in this study, WES data of all subjects was pseudonymised. This entailed the at random enciphering of patient DNA numbers to ascending numbers by a Genome Diagnostics member. In addition, clinical descriptions were condensed and classified according to the International Union of Immunological Societies (IUIS) classification (*Bousfiha et al., 2020*). Some of the included trios were part of previous publications: one was published as a clinical case report by D'hauw *et al.*, 19 were part of the IEI cohort of Arts *et al.*, and one was part of a study by Konrad *et al.* (*Figure 1—source data 1*; *Arts et al., 2019*; *Konrad et al., 2019*; *D'hauw et al., 2008*).

### Diagnostic whole exome sequencing

WES was performed as described earlier with minor modifications (*Lelieveld et al., 2016*). In brief, genomic DNA samples isolated from whole blood were processed at the Beijing Genomics Institute (BGI) Europe (BGI Europe, Copenhagen, Denmark) or the in-house sequencing facility. All samples were enriched for exonic DNA using Agilent (Agilent Technologies, Santa Clara, CA, United States) or Twist (Twist Bioscience, San Francisco, CA, United States) exome kits. DNA samples at BGI were sequenced on Illumina HiSeq4000 (Illumina Sequencing, San Diego, CA, United States) or DNBseq (MGI Tech, Shenzhen, China). In-house DNA samples were sequenced on Illumina NovaSeq6000 (Illumina Sequencing). Sequencing was performed with 2x100 base pair (DNBseq) or 2x150 base pair (HiSeq4000 and NovaSeq6000) paired-end sequencing reads. The average median sequence coverage was 124x with an average of 96% target coverage greater than 20x (*Figure 1—source data 1*).

Downstream processing was performed by an automated data analysis pipeline, including mapping of sequencing reads to the GRCh37/hg19 reference genome with the Burrows-Wheeler Aligner algorithm and Genome Analysis Toolkit variant calling and additional custom-made annotation (*Li and Durbin, 2010*; *McKenna et al., 2010*). The DeNovoCheck tool is part of the custom-made annotation and was used to align variants called in each member of the patient-parent trios, providing an indication whether variants were inherited or *de novo*. DNVs were filtered out if the variation reads in either parent exceeded 2% (*Lelieveld et al., 2016*; *de Ligt et al., 2012*). Subsequently, all single nucleotide variants (SNVs) or small insertion-deletions (indels) were annotated by a custom, in-house annotation pipeline. Copy number variants (CNVs) were assessed by the copy number inference from exome reads (CoNIFER) method, as of 2018 (*Krumm et al., 2012*).

Subsequently, variants in genes included in the *in silico* IEI panel were filtered to retain both inherited and *de novo* coding, non-synonymous variants with population frequencies below 1% in our

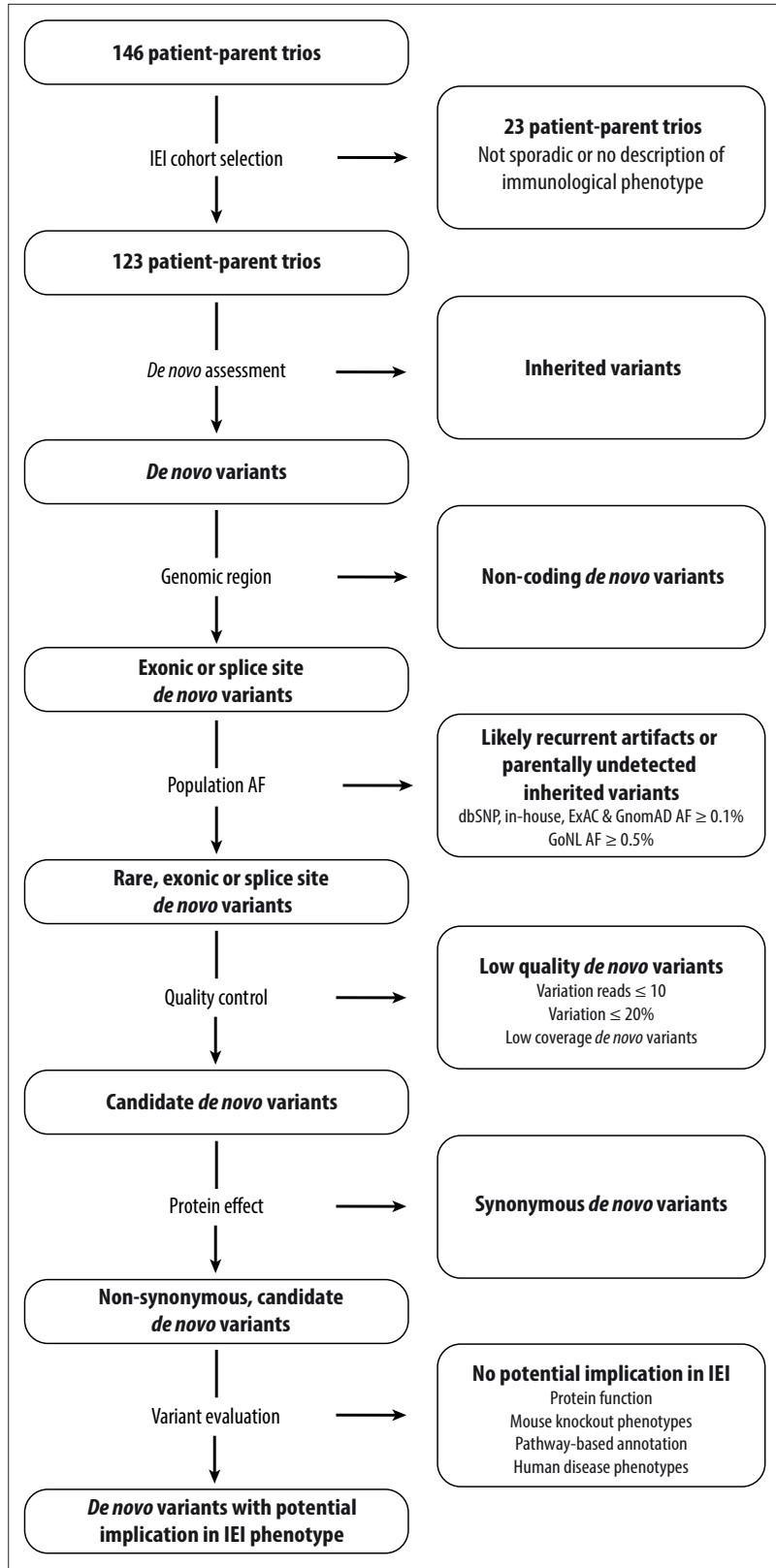

**Figure 1.** Schematic overview of patient inclusion, *de novo* variant filtering strategy and variant evaluation. Of the 146 eligible patient-parent trios, 123 trios met the inclusion criteria for this IEI cohort study. Whole exome sequencing data from these patient-parent trios was filtered to retain rare, non-synonymous candidate *de novo* variants in coding regions. Subsequently, variants were systematically evaluated at variant and gene level for their

*Figure 1 continued on next page*

*Figure 1 continued*

potential involvement in the patient's immunological phenotype. Abbreviations: IEI = inborn errors of immunity; dbSNP = Single Nucleotide Polymorphism Database; ExAC = Exome Aggregation Consortium; GnomAD = Genome Aggregation Database; AF = allele frequency; GoNL = Genome of the Netherlands.

The online version of this article includes the following source data, source code, and figure supplement(s) for figure 1:

**Source code 1.** R script for *de novo* variant filtering.

**Source data 1.** List of 123 patient-parent trios with patient characteristics and whole exome sequencing performance statistics.

**Source data 2.** List of all candidate rare, coding *de novo* variants found in the cohort of IEI patients.

**Source data 3.** *De novo* variant rate and distribution of *de novo* variant types across our IEI cohort in comparison to a reference cohort from *Kaplanis et al., 2020*.

**Figure supplement 1.** Distribution of rare, non-synonymous coding *de novo* variants among cases.

**Figure supplement 1—source data 1.** Number of candidate *de novo* variants per case.

in-house database or population databases (GnomAD and dbSNP) (*Karczewski et al., 2020*; *Sherry et al., 1999*). Variant prioritisation was performed by clinical laboratory geneticists of the Department of Human Genetics at the RUMC. SNVs, small indels or CNVs that were considered to be (partially) related to the phenotype were classified (five-tier classification) and reported according to guidelines of the Association for Clinical Genetic Science and the American College of Medical Genetics and Genomics (ACMG; *Richards et al., 2015*; *Wallis et al., 2013*). Variants that were denoted or classified as carriership of a variant in a known recessive disease gene, known risk factors or variants of uncertain significance or (likely) pathogenic variants in disease genes other than those associated with IEI and candidate variants in genes without any disease association, were additionally reported and are listed in *Table 1—source data 1*.

## *De novo* variant analysis

In this study, a research-based re-analysis was performed on 123 patient-parent trio WES datasets to assess the presence of candidate DNVs. For this, a standardised variant filtering strategy was scripted using R Studio version 3.6.2 (*Figure 1—source code 1*). Variants were filtered to retain rare (≤0.1% allele frequency in our in-house database and the population databases from Exome Aggregation Consortium (ExAC), Genome Aggregation Database (GnomAD) genomes and dbSNP as well as ≤0.5% in the Genome of the Netherlands (GoNL) database), coding, non-synonymous, possible DNVs, as annotated by the DeNovoCheck tool (*Figure 1*; *Lelieveld et al., 2016*; *de Ligt et al., 2012*; *Karczewski et al., 2020*; *Sherry et al., 1999*; *Lek et al., 2016*; *Boomsma et al., 2014*). Variants with ≤10 variation reads, ≤20% variant allele fraction or low coverage DNVs (*de Ligt et al., 2012*) were excluded. Moreover, synonymous SNVs and small indels were removed from the analysis. DNVs excluded by this filtering strategy were investigated for potential pathogenicity in known IEI genes. The remaining candidate DNVs are listed in *Figure 1—source data 2*. These DNVs were prioritised and systematically evaluated using variant and gene level metrics, encompassing database allele frequencies (including DNV counts in other datasets via denovo-db), nucleotide conservation, pathogenicity prediction scores, functional information and possible involvement in the immune system based on mouse knockout models, pathway-based annotation (i.e. Gene Ontology terms), and literature studies (*Karczewski et al., 2020*; *Wiel et al., 2019*; *Stephenson et al., 2019*; *Turner et al., 2017*). Prioritised candidate DNVs were visually inspected using the Integrative Genomics Viewer (IGV) and/or Alamut Visual Software version 2.13 (SOPHiA GENETICS, Saint Sulpice, Switzerland) to investigate biases that would give rise to false-positive variant calls. In addition, splice site DNVs were analysed using the SpliceAI prediction score (*Jaganathan et al., 2019*) and the Alamut Visual Software, which has incorporated splicing prediction tools such as SpliceSiteFinder-like, MaxEntScan, NNSPLICE, GeneSplicer and ESE tools.

## FBXW11 functional validation experiments

### Epstein–Barr virus (EBV)-B cell lines

Venous blood was drawn from patient 53 and collected in lithium heparin tubes. Epstein-Barr virus (EBV)-transformed B cell lines were created following established procedures (*Neitzel, 1986*). EBV-transformed lymphoblastoid cell lines (EBV-LCLs) from the patient and a healthy control were grown at 37 °C and 7.5% $CO_2$ in RPMI 1640 medium (Dutch Modification, Gibco; Thermo Fisher Scientific, Inc, Waltham, MA, United States) containing 15% foetal calf serum (FCS; Sigma-Aldrich, St Louis, MO, United States), 1% 10,000 U/µl penicillin and 10,000 µg/µl streptomycin (Sigma-Aldrich), and 2% HEPES (Sigma-Aldrich). The EBV-LCLs were cultured at a concentration of $10 \times 10^6$ in 150 cm$^2$ culture flasks (Corning, Corning, NY, United States) and treated with or without cycloheximide at 0.1% (20mL/20 mL medium; Sigma-Aldrich) for four hours. Cell pellets were then spun down, washed with PBS, snap-frozen in liquid nitrogen and stored at -80 °C.

## RNA splicing effect

RNA was isolated from the EBV-B cell pellets using the RNeasy Mini isolation kit (Qiagen, Hilden, Germany) according to the manufacturer's instructions. Subsequently, cDNA was synthesised from RNA with the iScript cDNA Synthesis Kit (Bio-Rad, Hercules, CA, United States). A primer set was designed (Primer3web, version 4.1.0) to span exon 11–13 of *FBXW11*, with the following sequences: Forward 5'-GAGAGCCGGAATCAGAGGTG-3'; Reverse 5'-GAATTGGTCCGATGCATCCG-3'. Subsequently, RT-PCR was performed using the AmpliTaq Gold 360 Master Mix (Life Technologies, Carlsbad, CA, United States). The amplified PCR products and Orange G ladder were electrophoresed on a 2% agarose gel with GelRed, and the resulting bands were cut out and analysed with Sanger sequencing.

## *Ex vivo* peripheral mononuclear blood cell (PBMC) experiments

Venous blood was drawn and collected in EDTA tubes. Immune cell isolation was conducted as described elsewhere (*Oosting et al., 2016*). In brief, PBMCs were obtained from blood by differential density centrifugation, diluted 1:1 in pyrogen-free saline over Cytiva Ficoll-Paque Plus (Sigma-Aldrich). Cells were washed twice in saline and suspended in cell culture medium (Roswell Park Memorial Institute (RPMI) 1640, Gibco) supplemented with gentamicin, 50 mg/mL; L-glutamine, 2 mM; and pyruvate, 1 mM. *Ex vivo* PBMC stimulations were performed with $5 \times 10^5$ cells/well in round-bottom 96-well plates (Greiner Bio-One, Kremsmünster, Austria) for 24 hr in the presence of 10% human pool serum at 37 °C and 5% carbon dioxide. For cytokine production measurements, cells were treated with *Candida albicans* yeast (UC820 heat-killed, $1 \times 10^6$ /mL), lipopolysaccharide (LPS, 10 ng/mL), *Staphylococcus aureus* (ATCC25923 heat-killed, $1 \times 10^6$ /mL) or TLR3 ligand Poly I:C (10 µg/mL) or left untreated in regular RPMI medium. After the incubation period and centrifugation, supernatants were collected and stored at -20 °C until the measurement using enzyme-linked immunosorbent assay (ELISA).

For flow cytometry experiments, PBMCs were cultured in U-bottom plates at a final concentration of $1 \times 10^6$ cells in 200 µL per well containing culture medium supplemented with 5% FCS (Sigma-Aldrich) at 37 °C and 5% carbon dioxide. Subsequently, cells were stimulated with phorbol 12-myristate 13-acetate (PMA, 12.5 ng/mL, Sigma-Aldrich) and ionomycin (500 ng/mL, Sigma-Aldrich) in duplicate for 30 min.

## Flow cytometry

PBMC suspensions were transferred to a V-bottom plate while pooling the duplicates. Following centrifugation for 2.5 min, cell surface markers were stained in the dark for 30 min at 4 °C with a monoclonal antibody mix containing anti-CD3-ECD (1:25; Beckman Coulter, Brea, CA, United States), anti-CD4-BV510 (1:50; BD Bioscience, Franklin Lakes, NJ, United States), anti-CD8-APC Alexa Fluor 700 (1:400; Beckman Coulter), and anti-CD14-FITC (1:50; Dako; Agilent Technologies). Subsequently, cells were washed twice with flow cytometry buffer (FCM buffer, 0.2% BSA in PBS) and fixed (BD Biosciences Cytofix, 554655) for 10 min at 37 °C. Next, cells were washed and permeabilised with perm buffer IV (1:10 diluted with PBS, BD Biosciences Phosflow, 560746) for 20 min on ice in the dark. Cells were then stained intracellularly with anti-NF-κB p65 (pS529)-PE antibody (1:50; eBioscience; Thermo Fisher Scientific, Inc, Waltham, MA, United States) for 20 min at 4 °C. After washing the cells twice in FCM-buffer, the suspensions were measured on a Beckman Coulter Navios EX Flow Cytometer using Navios System Software. Cell immunophenotypes were analysed using Kaluza Analysis Software

version 2.1 (Beckman Coulter). The mean fluorescent intensities (MFIs) were calculated using the median pNF-κB p65 expression levels within the gated immune cell populations of interest.

## Cytokine measurements

Levels of cytokines IL-1β, IL-6 and TNFα were determined in supernatants of stimulated PBMC cultures according to the instructions of the manufacturer (Duoset ELISA; R&D Systems, Minneapolis, MN, United States).

## Results

### Cohort characteristics

This retrospective cohort study systematically re-analysed patient-parent trio whole exome sequencing (WES) data of 123 patients with suspected, sporadic inborn errors of immunity (IEI) with the aim to identify candidate *de novo* single-nucleotide variants (SNVs) or small insertion-deletions (indels) (*Figure 1*). The selected IEI patients had a median age of 9 years (IQR 2–17), and two-thirds of the cases were below 18 years of age (*Table 1*). The sex distribution among patients was roughly equal. Classification of IEI phenotypes according to the International Union of Immunological Societies (IUIS) indicated that most cases presented with autoinflammatory syndromes, followed by immune dysregulation and combined, predominantly syndromal immunodeficiencies (*Bousfiha et al., 2020*). Eight patients remained unclassified due to limited clinical data.

### Reported genetic variants after diagnostic whole exome sequencing

Potential disease-causing SNVs and/or copy number variants (CNVs) were reported in 36 index patients after diagnostic WES (*Table 2*). Twenty-four patients were carriers of recessive disease alleles, previously characterised risk factors, variants of uncertain significance (VUS) or (likely) pathogenic variants affecting established disease genes other than those associated with IEI (*Table 2*). Of note, three of these patients carried *de novo* CNVs that met the diagnostic quality criteria. Patient 21 was diagnosed with an autoinflammatory disorder and carried a 17q terminal deletion of uncertain clinical significance. Overlapping CNVs have been previously reported in DECIPHER (*Firth et al., 2009*). A patient with non-syndromal combined immunodeficiency carried a *de novo* CNV involved in the Chromosome

**Table 1.** Patient cohort characteristics. Demographic and phenotypic characteristics of the 123 patients with suspected inborn errors of immunity that were included in the study.

| Characteristic | Total N=123 |
|---|---|
| Demographics | |
| Age*, median (IQR) y | 9 (2-17) |
| <18 y, % | 67.4 |
| >18 y, % | 33.6 |
| Sex ratio, M:F | 50.4:49.6 |
| Distribution of clinical phenotypes † | |
| Severe combined immunodeficiency, n (%) | 9 (7.3) |
| Suspected SCID (low TRECs), n | 5 |
| Other, n | 4 |
| Combined immunodeficiency, n (%) | 22 (17.9) |
| Syndromal, n | 20 |
| Non-syndromal, n | 2 |
| Primary antibody deficiency, n (%) | 14 (11.4) |
| CVID, n | 14 |
| Agammaglobulinemia, n | 0 |
| Other, n | 0 |
| Immune dysregulation, n (%) | 20 (16.3) |
| HLH/EBV, n | 5 |
| Autoimmunity, n | 15 |
| Autoinflammatory syndrome, n (%) | 22 (17.9) |
| Periodic fever syndrome, n | 19 |
| Interferonopathy, n | 0 |
| Other, n | 3 |
| Phagocyte defect, n (%) | 5 (4.1) |
| Functional defect, n | 1 |
| Neutropenia/other, n | 4 |
| Innate/intrinsic immune defect, n (%) | 16 (13.0) |
| Bacterial/parasitic, n | 2 |
| MSMD/Viral, n | 7 |
| Other, n | 7 |
| Complement deficiencies, n (%) | 0 (0.0) |
| Bone marrow failure, n (%) | 10 (8.1) |
| Phenocopies of PIDs, n (%) | 0 (0.0) |
| Unclassified, n (%) | 5 (4.1) |

*Table 1 continued on next page*

*Table 1 continued*

| Characteristic | Total N=123 |
| --- | --- |

Abbreviations: IQR = interquartile range; SCID = severe combined immunodeficiency; TREC = T cell receptor excision circle; CVID = common variable immunodeficiency; HLH = haemophagocytic lymphohistiocytosis; EBV = Epstein-Barr virus; MSMD = Mendelian susceptibility to mycobacterial disease; PID = primary immunodeficiency.

*The age at the time of genetic testing is indicated, since the age of onset has not been documented for all cases.

†Categorization of phenotypes is based on the IUIS classification of 2019 (14).

The online version of this article includes the following source data for table 1:

**Source data 1.** List of patient-parent trios with variants identified in genes outside the diagnostic IEI gene panel, or classified as risk factors, carriership or variants of uncertain significance.

22q11.2 microduplication syndrome (patient 69, OMIM #608363). Another *de novo* CNV was identified in patient 115 who was diagnosed with severe combined immunodeficiency. This young female carried a large duplication in Xq of uncertain clinical significance.

In 12 patients, (likely) pathogenic SNVs were identified in known IEI genes that (partially) explain the patient's immunological phenotype (*Table 2*, details shown in *Table 3*). While most variants were inherited, one patient with Muckle-Wells syndrome (patient 59) carried a *de novo* missense variant in *NLRP3* (NM_001079821.2:c.1049C>T p. (Thr350Met)). This variant has previously been described in patients with Muckle-Wells syndrome (*Dodé et al., 2002*; *Jiménez-Treviño et al., 2013*). Consequently, the *NLRP3 de novo* variant (DNV) was classified as pathogenic (*Richards et al., 2015*; *Wallis et al., 2013*).

Overall, routine diagnostic WES analysis provided a likely molecular diagnosis for (part) of the phenotype in 18 patients (14.6%) based on established mutational mechanisms and disease associations (*Table 2*).

## Rare, non-synonymous *de novo* variants in novel IEI candidate genes

Next, re-analysis was performed on WES data of all 123 sporadic IEI cases and their parents to systematically identify and interpret DNVs in novel IEI genes. Automated DNV filtering retained a total of 172 candidate DNVs that were rare (148 DNVs were absent from GnomAD genomes) and located in either exonic or splice site regions (the complete list can be found in *Figure 1—source data 2*). The total number of candidate DNVs among patients ranged between zero and six (*Figure 1—figure supplement 1*). Moreover, the average number of candidate DNVs was comparable to recent literature (*Figure 1—source data 3*). Of these candidate DNVs, 124 were non-synonymous and therefore expected to exert an effect on protein function (*Figure 1—source data 2*). Two pairs of patients carried candidate DNVs in the same gene, *GIGYF1* (patients 49 and 83) and *MAP3K10* (patients 98 and 118). However, these patients did not share phenotypic features and the function of the proteins encoded by these genes could not be linked to the respective patient phenotype.

Subsequently, all non-synonymous candidate DNVs were systematically evaluated based on information on variant and gene level metrics, leading to the selection of 14 candidate DNVs potentially causing IEI (*Tables 3 and 4*), including the above-mentioned variant in the known IEI gene

**Table 2.** Genetic findings after routine diagnostic panel analysis.
Genetic variants reported after routine diagnostic whole exome sequencing analysis of the 123 patients included in this cohort of inborn errors of immunity.

| Total cases in which a genetic variant was reported, n (%) | 36 (29.3) | Patient nr. |
| --- | --- | --- |
| (Likely) pathogenic mutation, n (%) | 18 (14.6) | |
| Within IEI gene panel, n (%) | 12 (9.8) | All patients listed in *Table 3* |
| Beyond IEI gene panel, n (%) | 6 (4.9) | 1, 3, 40, 69, 85, 103 (*Table 1—source data 1*) |
| Other variants, n (%) | 19 (15.4) | *Table 1—source data 1* |
| Risk factor, n (%) | 6 (4.9) | 21, 44, 55, 56, 68, 112 |
| Carriership recessive allele, n (%) | 7 (5.7) | 3, 7, 16, 23, 32, 44, 76 |
| Variant of unknown significance, n (%) | 9 (7.3) | 6, 21, 23, 45, 54, 80, 100, 101, 115 |

**Table 3.** Patients with previously reported single nucleotide variants, small insertion-deletions, or copy number variants that may (partially) explain the patient's immunological phenotype.

Listed variants were identified prior to the research-based systematic re-analysis of the current study following diagnostic gene panel analysis for inborn errors of immunity.

| Patient nr. | Sex | Age range at sampling | Phenotype (IUIS classification) | Variant | Mutational mechanism | ACMG classification | ClinVar accession | Comments |
|---|---|---|---|---|---|---|---|---|
| 10 | F | 0–5 | Immune dysregulation, HLH/EBV | AP3B1 Chr5(GRCh37):g.77563371del NM_003664.4:c.177del p.(Lys59fs) | | Pathogenic | VCV000224763 | Hermansky-Pudlak syndrome 2 (OMIM #608233) |
| | | | | AP3B1 Chr5(GRCh37):g.77423980_77423983del NM_003664.4:c.1839_1842del p.(Asp613fs) | AR (ch) LoF | Pathogenic | VCV000224764 | |
| | | | | FAS Chr10(GRCh37):g.90774167_90774186dup NM_000043.6:c.968_987dup p.(Glu330fs) | AD (htz) LoF | Pathogenic | VCV000016509 | Autoimmune lymphoproliferative syndrome, type IA (OMIM #601859) |
| 12 | F | 11–15 | CID, syndromal | seq(GRCh37) del(t16)(p11.2p11.2) NC_000016.9:g.(29469093_29624260)_(30199846_30208282)del | AD (htz) LoF | Pathogenic | - | 16 p11.2 deletion syndrome (OMIM #611913) |
| 26 | F | 0–5 | Bone marrow failure | DHFR Chr5(GRCh37):g.79950248C>T NM_000791.3:c.61G>A p.(Gly21Arg) | AR (hmz) LoF | Likely pathogenic | - | Megaloblastic anaemia due to dihydrofolate reductase deficiency (OMIM #613839) Affected sibling carries equal variant |
| 59 | M | 6–10 | Autoinflammatory disorder | NLRP3 Chr1(GRCh37):g.247587794C>T NM_001079821.2:c.1049C>T p.(Thr350Met) | AD (htz) LoF | Pathogenic | - | Muckle-Wells syndrome (OMIM #191900) De novo SNV |
| 61 | M | 0–5 | CID, syndromal | MKL1 Chr22(GRCh37):g.40815086dup NM_020831.4:c.1356dup p.(Val453Argfs) | AR (hmz) LoF | Likely pathogenic | - | Immunodeficiency 66 (OMIM #61847) Affected sibling carries equal variant |
| 77 | F | 0–5 | CID, syndromal | ALOXE3 Chr17(GRCh37):g.8006708G>A NM_021628.2:c.1889C>T p.(Pro630Leu) | AR (hmz) LoF | Pathogenic | - | Congenital ichthyosis 3 (OMIM #606545) |
| 91 | F | 0–5 | Suspected SCID (low TRECs) | FOXN1 Chr17(GRCh37):g.26857765A>G NM_003593.2:c.831-2A>G p.? | AD (htz) LoF | Likely pathogenic | - | T-cell lymphopenia, infantile, with or without nail dystrophy (OMIM #618806) |
| | | | | CD55 Chr1(GRCh37):g.207497984dup NM_001300902.1:c.367dup p.(Thr123fs) | AR (hmz) LoF | Pathogenic | - | Complement hyperactivation, angiopathic thrombosis, and protein-losing enteropathy (OMIM #226300) |
| 102 | F | 11–15 | Immune dysregulation, autoimmunity and others | PET117 Chr20(GRCh37):g.18122927C>T NM_001116481.1:c.172C>T p.(Gln58*) | AR (hmz) LoF | Likely pathogenic | VCV000981504 | Mitochondrial complex IV deficiency, nuclear type 19 (OMIM #619063) |
| 105 | M | 31–35 | Defects in intrinsic and innate immunity, MSMD and viral infection | TLR7 ChrX(GRCh37):g.12905756_12905759del NM_016562.3:c.2129_2132del p.(Gln710fs) | XLR (hemi) LoF | Pathogenic | VCV000977232 | Immunodeficiency 74, COVID19-related (OMIM #301051) Affected sibling carries equal variant |
| 114 | M | 6–10 | Immune dysregulation, autoimmunity and others | LRBA Chr4(GRCh37):g.151835415del NM_006726.4:c.1093del p.(Tyr365fs) | AR (hmz) LoF | Pathogenic | - | Common variable immunodeficiency 8 (OMIM #614700) |
| 120 | M | 11–15 | Congenital defect of phagocyte, functional defects | NCF1 Chr7(GRCh37):g.74191615_74191616del NM_000265.5:c.75_76del p.(Tyr26fs) | AR (hmz) LoF | Pathogenic | VCV000002249 | Chronic granulomatous disease 1 (OMIM #233700) |
| 122 | M | 0–5 | Suspected SCID (low TRECs) | FOXN1 Chr17(GRCh37):g.26851540del NM_003593.2.1:c.143del p.(Cys48fs) | AD (htz) LoF | Pathogenic | - | T-cell lymphopenia, infantile, with or without nail dystrophy (OMIM #618806) |

Abbreviations: IUIS = International Union of Immunological societies; ACMG = American College of Medical Genetics and Genomics; HLH = haemophagocytic lymphohistiocytosis; EBV = Epstein-Barr virus; OMIM = Online Mendelian Inheritance in Man; (S)CID = (severe) combined immunodeficiency; TREC = T cell receptor excision circle; MSMD = Mendelian susceptibility to mycobacterial disease; AR = autosomal recessive; AD = autosomal dominant; XLR = X-linked recessive; ch = compound heterozygous; htz = heterozygous; hmz = homozygous; hemi = hemizygous; LoF = loss-of-function; SNV = single nucleotide variant.

**Table 4.** Identification of 13 heterozygous, rare and non-synonymous candidate *de novo* variants. The 124 non-synonymous candidate *de novo* variants were systematically evaluated based on the potential to be damaging to gene and protein function and the possible involvement in the patient's immunological phenotype.

| Patient nr. | Sex | Age range at sampling | Phenotype (IUIS classification) | De novo variant | GnomAD AF in % | in-house AF in % | PhyloP | CADD | VarMap | MetaDome | Coding DNV in denovo-db (protein effect) | LOEUF | Function | Literature | Comments |
|---|---|---|---|---|---|---|---|---|---|---|---|---|---|---|---|
| **Missense SNVs** | | | | | | | | | | | | | | | |
| 1 | M | 11–15 | SCID | PSMB10 Chr16(GRCh37): g.67968809C>T NM_002801.3: c.601G>A p.(Gly201Arg) | 0 | 0 | 5 | 32 | Likely deleterious | Neutral | - | 1.37 | Immuno- and thymoproteasome subunit | Homozygous *Psmb10* variant in mice causes SCID and systemic autoinflammation (*Treise et al., 2018*). Homozygous *PSMB10* variant in humans cause PRAAS, no immunodeficiency (*Sarrabay et al., 2020*). | Revertant somatic mosaicism (VAF: 39.7%). Additional inherited SNV and partial somatic UPD16 (*Table 1—source data 1*). |
| 9 | M | 6–10 | Predominantly antibody deficiency, hypogamma-globulinemia | RPL27A Chr11(GRCh37): g.8707228T>C NM_000990.4: c.322T>C p.(Tyr108His) | 0.0032 | 0.0041 | 7.4 | 27.4 | Likely deleterious | Intolerant | - | 0.39 | Ribosomal subunit | Ribosomopathies may include immunological defects (*Khan et al., 2011*). | |
| 27 | M | 11–15 | Autoinflammatory disorder | TAOK2 Chr16(GRCh37): g.29997683C>T NM_016151.3: c.2090C>T p.(Ala697Val) | 0 | 0 | 4.8 | 22.5 | Possibly deleterious | Slightly intolerant | 6 (4 mis) | 0.24 | Serine/threonine-protein kinase (p38 MAPK pathway) | Homozygous TAOK2 variant causes abnormal T cell activation in two patients with inflammatory bowel disease (*Molho-Pessach et al., 2017*). | |
| 28 | F | 16–20 | Predominantly antibody deficiency, hypogamma-globulinemia | KCTD9 Chr8(GRCh37): g.25292997C>T NM_017634.3: c.695G>A p.(Arg232His) | 0 | 0.0082 | 5.8 | 32 | Likely deleterious | Intolerant | - | 0.52 | Substrate-specific adapter | Involved in NK cell activation (*Chen et al., 2013*). | |
| 52 | M | 11–15 | Predominantly antibody deficiency, hypogamma-globulinemia | SCRIB Chr8(GRCh37): g.144874432C>T NM_182706.4: c.4472G>A p.(Arg1491Gln) | 0.0032 | 0 | 4.2 | 29.9 | Possibly deleterious | Intolerant | 5 (4 mis) | 0.31 | Scaffold protein | Involved in uropod and immunological synapse formation, and ROS production by antigen-presenting cells (*Barreda et al., 2020*). | |
| 58 | F | 21–25 | Unclassified | CTCF Chr16(GRCh37): g.67645905G>T NM_006565.4: c.833G>T p.(Arg278Leu) | 0 | 0 | 9.7 | 24.7 | Possibly deleterious | Highly intolerant | 12 (11 mis) | 0.15 | Transcriptional insulator | CTCF variants cause neurodevelopmental disorders, sometimes associated with recurrent infections and minor facial dysmorphisms (*Konrad et al., 2019*). | Published (*Konrad et al., 2019*). |
| 75 | F | 6–10 | Bone marrow failure | FUBP1 Chr1(GRCh37): g.78435621A>C NM_001303433.1: c.199T>G p.(Leu67Val) | 0 | 0 | 2.6 | 24.8 | Possibly deleterious | Intolerant | 1 (0 mis) | 0.12 | Transcriptional regulator that binds FUSE upstream of the c-myc promoter | Essential for long-term repopulating hematopoietic stem cell renewal (*Rabenhorst et al., 2015*). Fubp1 KO mice show cerebral hyperplasia, pulmonary hypoplasia, pale livers, hypoplastic spleen, thymus, and bone marrow, cardiac hypertrophy, placental distress, and small size (*Zhou et al., 2016*). | |
| 118 | F | 0–5 | Immune dysregulation, autoimmunity and others | RUNX3 Chr1(GRCh37): g.25256227C>T NM_004350.2: c.133G>A p.(Gly45Arg) | 0 | 0 | 2.4 | 18 | Possibly deleterious | Slightly tolerant | 1 (1 mis) | 0.42 | Transcriptional regulator | RUNX3 regulates CD8+T cell thymocyte development, maturation of cytotoxic CD8+T cells and the function of innate lymphoid cells 3 via stimulation of RORγt (*Ebihara et al., 2015*). Runx3 KO mice spontaneously develop inflammatory bowel disease and gastric lesions (*Brenner et al., 2004*). | |
| **Frameshift SNVs** | | | | | | | | | | | | | | | |
| 49 | M | 26–30 | Predominantly antibody deficiency, hypogamma-globulinemia | DDX1 Chr2(GRCh37): g.15769802dup NM_004939.2: c.1952dup p.(Trp652fs) | 0 | 0 | - | - | - | - | 4 (0 fs) | 0.28 | RNA helicase | Part of a dsRNA sensor that activates the NF-κB pathway and type I interferon responses (*Zhang et al., 2011*). | |

*Table 4 continued on next page*

*Table 4 continued*

| Patient nr. | Sex | Age range at sampling | Phenotype (IUIS classification) | De novo variant | GnomAD AF in % | in-house AF in % | PhyloP | CADD | VarMap | MetaDome | Coding DNV in denovo-db (protein effect) | LOEUF | Function | Literature | Comments |
|---|---|---|---|---|---|---|---|---|---|---|---|---|---|---|---|
| 78 | F | 6–10 | CID, syndromal | KMT2C Chr7(GRCh37): g.151860074del NM_170606.2: c.10588del p.(Ser3530Leufs*3) | 0 | 0 | - | - | - | - | 19 (4 fs) | 0.12 | Histone methyltransferase | KMT2C de novo variant causes Kleefstra syndrome 2, sometimes associated with recurrent respiratory infections (*Koemans et al., 2017*). | |

**Small in-frame indel**

| Patient nr. | Sex | Age range at sampling | Phenotype (IUIS classification) | De novo variant | GnomAD AF in % | in-house AF in % | PhyloP | CADD | VarMap | MetaDome | Coding DNV in denovo-db (protein effect) | LOEUF | Function | Literature | Comments |
|---|---|---|---|---|---|---|---|---|---|---|---|---|---|---|---|
| 108 | M | 21–25 | Bone marrow failure | NSD2 Chr4(GRCh37): g.1959681_1959687delins TTTTTCT NM_133330.2: c.2903_2909delins TTTTTCT p.(Arg968_Arg970delinsLeuPheLeu) | - | - | - | - | - | - | - | 0.12 | Histone methyltransferase | NSD2 de novo LoF variant causes mild Wolf-Hirschhorn syndrome (*Barrie et al., 2019*). Unclear role in immunity. | Postzygotic mosaicism (VAF 29%). |

**Splice site SNVs**

| Patient nr. | Sex | Age range at sampling | Phenotype (IUIS classification) | De novo variant | GnomAD AF in % | in-house AF in % | PhyloP | CADD | SpliceAI Acceptor Gain | SpliceAI Acceptor Loss | Coding DNV in denovo-db (protein effect) | LOEUF | Function | Literature | Comments |
|---|---|---|---|---|---|---|---|---|---|---|---|---|---|---|---|
| 53 | F | 11–15 | Autoinflammatory disorder | FBXW11 Chr5(GRCh37): g.171295802T>C NM_012300.2: c.1468-2A>G p.? | 0 | 0 | 7.9 | 34 | 0.0134 | 0.9862 | 2 (0 ss) | 0.31 | Component of SCF (SKP1-CUL1-F-box) E3 ubiquitin ligase complex | Involved in the regulation of NF-κB signalling (*Wang et al., 2018*). | |
| 119 | F | 11–15 | Autoinflammatory disorder | RELA Chr11(GRCh37): g.65423234C>T NM_021975.3: c.959-1G>A p.? | 0 | 0 | 3.5 | 34 | 0.7968 | 0.9991 | - | 0.18 | Transcription factor p65 (NF-κB subunit) | Heterozygous RELA variant causes chronic mucocutaneous ulceration (*Badran et al., 2017*). | |

Abbreviations: IUIS = International Union of Immunological Societies; GnomAD = Genome Aggregation Database; AF = allele frequency; CADD = Combined Annotation Dependent Depletion; DNV = de novo variant; LOEUF = loss-of-function observed/expected upper bound fraction; SNV = single nucleotide variant; indel = insertion-deletion; (S)CID = severe combined immunodeficiency; NA = not applicable; mis = missense; fs = frameshift; ss = splice site; MAPK = mitogen-activated protein kinase; FUSE = far upstream element; NK = natural killer; ROS = reactive oxygen species; KO = knockout; dsRNA = double-stranded RNA; NF-κB = nuclear factor kappa-light-chain-enhancer of activated B cells; LoF = loss-of-function; VAF = variant allele fraction; UPD16 = uniparental disomy of chromosome 16.

*NLRP3*. The 13 novel IEI candidate DNVs were found in patients with different IEI phenotypes, although three subtypes reoccurred: predominantly antibody deficiency (hypogammaglobulinemia), autoinflammatory disorder and bone marrow failure. Candidate DNVs that were considered most promising based on variant and gene level metrics are presented in more detail in the following paragraphs.

A patient with an autoinflammatory phenotype characterised by mucocutaneous ulceration of mouth and genital area carried a DNV in *RELA* that was located in the canonical splice acceptor site preceding exon 10 (patient 119, *Table 4*). The guanine to adenine change was predicted to compromise the splice acceptor site by transferring it to the first guanine of exon 10, leading to an out-of-frame exon. The resulting frameshift was therefore assumed to cause a reduction in functional RelA protein by nonsense-mediated decay. RelA is also known as p65 and is critically involved in nuclear factor kappa-light-chain-enhancer of activated B cells (NF-κB) heterodimer formation and consequent activation of NF-κB-mediated proinflammatory signalling. Although *RELA* has already been reported as an IEI gene in a previous IUIS classification (*Tangye et al., 2020*), it was not yet listed in the IEI *in silico* gene panel of our Department of Human Genetics (*Radboudumc, 2021*), because evidence was considered insufficient at the time.

In addition, a private *de novo* missense variant in *PSMB10* was found in a patient with Omenn syndrome with severe combined immunodeficiency (SCID), ectodermal dysplasia, alopecia, hypodontia and anonychia (patient 1, *Table 4*). The clinical phenotype of this patient has been previously reported (*D'hauw et al., 2008*). The DNV was predicted to be pathogenic based on the majority of variant and gene level metrics. In additional data that was available from a single-nucleotide polymorphism (SNP) micro-array, it was shown that the genomic location of *PSMB10* was spanned by a partial somatic uniparental disomy of chromosome 16 (UPD16) (manuscript in preparation). *PSMB10* encodes the β2i-subunit of the immuno- and thymoproteasome and mutations leading to a loss of PSMB10 protein function have been associated with severe immunological defects (*Treise et al., 2018*; *Sarrabay et al., 2020*). Furthermore, another candidate DNV was identified in a patient with common variable immunodeficiency (CVID) due to a B cell maturation defect, auto-immune cytopenia, polyclonal T cell large granular lymphocytes in the bone marrow, recurrent viral infections, psoriasis and alopecia areata, (patient 49, *Table 4*). This frameshift variant in *DDX1* was predicted to cause loss of protein function. *DDX1* encodes a RNA helicase, which is part of a double-stranded RNA sensor that activates the NF-κB pathway and type I interferon responses (*Zhang et al., 2011*). Moreover, DDX1 is involved in the regulation of hematopoietic stem and progenitor cell homeostasis (*Zhang et al., 2011*; *Wang et al., 2021*).

Another frameshift DNV in *KMT2C* was carried by a patient with a syndromal combined immunodeficiency characterised by recurrent ear infections, developmental delay, low-average intelligence level and facial dysmorphism (patient 78, *Table 4*). The variant was predicted to lead to a loss-of-function (LoF) of the KMT2C protein, which acts as a histone methyltransferase in the regulation of chromatin organisation.

Lastly, a DNV affecting *FBXW11* was identified in a patient with an autoinflammatory disorder characterised by recurrent periodic fever and severe headaches (patient 53, *Table 4*). *FBXW11* encodes a component of SCF (SKP1-CUL1-F-box) E3 ubiquitin ligase complex, TrCP2, that is involved in the regulation of NF-κB signalling through the ubiquitination of several of its components (*Wang et al., 2018*; *Kanarek and Ben-Neriah, 2012*). An important function of both the TrCP1 and TrCP2 isoforms is the regulation of IκBα degradation, leading to subsequent activation of NF-κB and release of proinflammatory cytokines (*Kim et al., 2015*; *Yaron et al., 1998*). The identified DNV affected the canonical splice acceptor site preceding exon 12 (NM_012300.2:c.1468-2A>G) and was predicted to lead to skipping of exon 12 based on splicing prediction by the Alamut Visual Software and to be deleterious by all utilised *in silico* prediction tools. The predicted RNA splicing defect leading to an in-frame, shortened RNA transcript was confirmed in Epstein-Barr virus (EBV) transformed B cells from the patient (*Figure 2—figure supplement 1*).

The other candidate DNVs will not be described in detail here, as there is insufficient evidence to suggest pathogenicity or a genotype-phenotype relationship. Future discovery of cases with DNVs in the presented genes and overlapping clinical phenotypes could encourage further in-depth research into the possible mutational mechanisms.

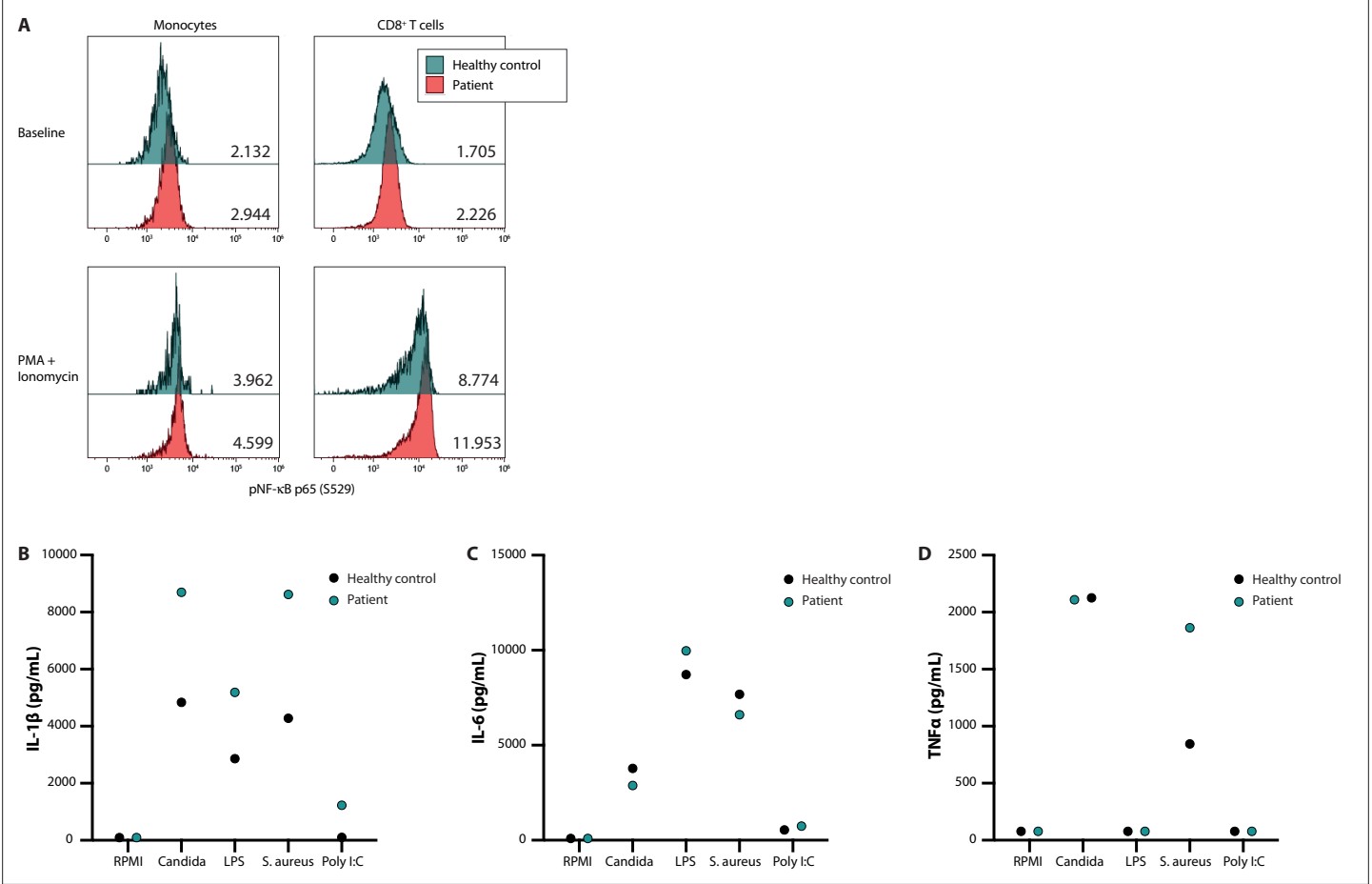

**Figure 2.** NF-κB signalling and production of innate cytokines upon *ex vivo* PBMC stimulation. Panel A shows the median fluorescence intensity expression levels of pNF-κB p65 (S529) in peripheral blood CD14 +monocytes and CD8 +T cells from a healthy control (blue) and patient 53 (red), in the absence (baseline) or presence of phorbol 12-myristate 13-acetate and ionomycin stimulation, with the absolute values indicated in the lower right corner. Panels B, C, and D display the production of IL-1β, IL-6, and TNFα, respectively, after *ex vivo* stimulation for 24 hr.

The online version of this article includes the following source data and figure supplement(s) for figure 2:

**Source data 1.** Original vector file of *Figure 2*.

**Source data 2.** Uncropped gel with labels.

**Source data 3.** Raw data of cytokine measurements.

**Figure supplement 1.** RNA splicing effect of the FBXW11 *de novo* splice site variant (c.1468-2A>G).

## Functional validation of *FBXW11 de novo* variant

In addition to systematic DNV analysis, we have selected the candidate DNV in *FBXW11* for functional validation as part of this study to provide further evidence for a causal genotype-phenotype relationship (patient 53, *Table 4*). As such, the putative effects on NF-κB signalling and the downstream production of pro-inflammatory cytokines were investigated in peripheral blood mononuclear cells (PBMC) extracted from the patient and a healthy control. In unstimulated PBMC of the patient showed higher levels of phosphorylated NF-kB p65 compared to the control. *Ex vivo* stimulation of these PBMC with phorbol 12-myristate 13-acetate (PMA) and ionomycin for 30 min led to higher NF-κB activation, reflected by p65 phosphorylation fluorescence intensity measured by flow cytometry, as compared to the healthy control (*Figure 2*, panel A). The greatest difference was observed in the lymphocyte subset, particularly in CD8 +T cells (*Figure 2*, panel A). Subsequently, the downstream production of the cytokines IL-1β, IL-6, and TNFα was investigated. The patient-derived PBMC produced more IL-1β upon *in vitro* stimulation with the heat-killed pathogens *Candida albicans* and *Staphylococcus aureus*, the TLR4 agonist lipopolysaccharides (LPS) and the TLR3 ligand Poly I:C after

24 hr, as compared to the healthy control (*Figure 2*, panel B). This trend was not observed for the production of IL-6 and TNFα (*Figure 2*, panels C and D). These results indicate that the *FBXW11* DNV leads to a splicing defect with skipping of exon 12, resulting in a shorter transcript and increased NF-κB signalling and downstream IL-1β production.

## Discussion

We investigated the potential benefit of trio-based whole exome sequencing (WES) over routine single WES analysis in a retrospective cohort of 123 patients with suspected, sporadic inborn errors of immunity (IEI). Systematic analysis of *de novo* SNVs and small insertion-deletions (indels) led to the identification of 14 candidate *de novo* variants (DNVs), of which two were in known IEI genes and classified as pathogenic (*NLRP3, RELA*). Of the 12 variants in potentially novel candidate genes for IEI, four were considered to be most likely pathogenic (*PSMB10, DDX1, KMT2C, FBXW11*) based on gene and variant level metrics. Additionally, we have provided functional evidence that the *FBXW11* splice site DNV led to skipping of exon 12 resulting in the transcription of an altered protein product and subsequent downstream activation of NF-κB signalling with higher IL-1β production capacity.

We have performed a systematic DNV analysis in patients with a suspected, sporadic IEI. On average, these patients carried 1.4 DNVs in coding regions, a rate comparable to other, larger studies, indicating that DNV enrichment or depletion in IEI patients is unlikely (*Kaplanis et al., 2020*). Based on gene and variant level information, 14 DNVs (11.4%) were considered potential disease-causing candidates. Six of the candidate DNVs (4.9%) were considered likely or possibly pathogenic variants, while the consequence of the other eight DNVs (6.5%) was uncertain.

Two DNVs were in IEI genes (*NLRP3, RELA*) listed in the most recent IUIS classification and were classified as pathogenic (*Richards et al., 2015*; *Wallis et al., 2013*; *Tangye et al., 2020*). The hetero-zygous *NLRP3* variant in patient 59 (p.Thr350Met) with Muckle-Wells syndrome had been reported in patients with a similar phenotype (*Dodé et al., 2002*; *Jiménez-Treviño et al., 2013*). Moreover, the canonical splice site DNV affecting *RELA* in patient 119 with mucocutaneous ulceration was predicted to lead to a loss of the splice acceptor site and a subsequent frameshift. Heterozygous loss-of-function (LoF) mutations causing RelA haploinsufficiency have been reported as a cause of chronic mucocu-taneous ulceration and familial Behçet's disease (*Badran et al., 2017*; *Adeeb et al., 2021*). Badran *et al.* reported a family of four affected family members with mucocutaneous ulceration harbouring a mutation in the canonical donor splice site of exon 6 (NM_021975:c.559+1 G>A), likely leading to a premature stop codon and haploinsufficiency (*Badran et al., 2017*). The DNV in *RELA* was not picked up in our diagnostic *in silico* IEI gene panel (*Radboudumc, 2021*), because evidence was considered insufficient at the time. Based on the matching phenotype and similar mutational mechanism this DNV has now been classified as pathogenic, which could potentially carry implications for therapy with anti-tumour necrosis factor alpha (TNFα) inhibitors (*Adeeb et al., 2021*).

Moreover, DNVs in the potentially novel IEI genes *PSMB10, DDX1, KMT2C,* and *FBXW11* were considered the most promising candidate DNVs based on the predicted variant effect and immuno-logical function of the respective gene. The private missense DNV in *PSMB10* was found in a patient with clinically diagnosed Omenn syndrome and showed high scores for pathogenicity. The presumed deleterious effect was further supported by the extremely rare occurrence of revertant mosaicism in this patient (unpublished data), that is, somatic and recurrent uniparental disomy 16q overlapping the *PSMB10* locus, suggesting a strong (cellular) effect of this variant. A homozygous missense variant in *PSMB10* has been shown previously in a 3-year-old Algerian female with autoinflammatory signs suggestive of proteasome-associated autoinflammatory syndrome (PRAAS), leading to disturbed formation of the 20S proteasome (*Sarrabay et al., 2020*). In addition, it has been shown in mice that another homozygous *PSMB10* variant (p.Gly170Trp) could induce severe combined immunodeficiency (SCID) and systemic autoinflammation, while heterozygous mice only had a T cell defect (*Treise et al., 2018*). PRAAS is predominantly caused by autosomal recessive or digenic heterozygous mutations in proteasome subunit genes or their chaperone proteins, although heterozygous (*de novo*) mutations have also been shown to underlie PRAAS (*Sarrabay et al., 2020*; *Agarwal et al., 2010*; *Brehm et al., 2015*). A *de novo* missense variant in the b1i-subunit *PSMB9* (NM_002800; c.467 G>A; p.G156D) was found in three unrelated infants with a type I interferonopathy with immunodeficiency (PRAAS-ID). This variant resulted in impaired maturation and activity of the immunoproteasome in patient-derived B lymphoblastoid cell lines (*Kanazawa et al., 2021*; *Kataoka et al., 2021*), a phenotype that was

mirrored in mice (*Kanazawa et al., 2021*). It is interesting to consider that the *PSMB10* DNV could act through a similar autosomal dominant mutational mechanism by affecting the formation of the immunoproteasome, as has been shown in mice harbouring a mutation affecting an amino acid in close proximity to the identified DNV in the patient (*Treise et al., 2018*). However, the functional consequence and pathogenic relevance of the candidate DNV in *PSMB10* remain to be confirmed.

In addition, a *de novo* frameshift variant in the highly intolerant *DDX1* (pLI 0.994) was identified in a patient with hypogammaglobulinemia, hematopoietic cell lineage abnormalities and recurrent infections. Although DDX1 plays a role in NF-κB signalling, type I interferon responses and the regulation of hematopoietic stem and progenitor cell homeostasis (*Zhang et al., 2011*; *Wang et al., 2021*), a causal genotype-phenotype relationship remains unclear. Furthermore, the *de novo* frameshift variant in *KMT2C* was detected in a patient with combined immunodeficiency and a neurodevelopmental phenotype, displaying partial phenotypic overlap with Kleefstra syndrome type 2 that has already been associated with *de novo* LoF mutations in *KMT2C* (*Koemans et al., 2017*). Two of the six individuals described by Koemans *et al.* were reported to have recurrent respiratory infections (*Koemans et al., 2017*). The occurrence of immunological symptoms in patients with mutations in chromatin-regulating genes is increasingly being recognised in the field of intellectual disability (ID) (*Ehrlich et al., 2008*; *Hoffman et al., 2005*). Therefore, more in-depth characterisation of patients with *KMT2C* mutations and predominant ID phenotypes might indicate (mild) immunological phenotypes that overlap with the phenotype of our patient, in support of pathogenicity of the observed DNV.

Another candidate DNV in a potentially novel IEI gene was identified in the highly conserved *FBXW11* (pLI 0.976). This DNV affected the canonical splice acceptor site preceding exon 12 and was shown to create a splice defect leading to exon skipping with a shortened transcript that retained expression at the RNA level. Exon 12 encodes component 7 of the WD40 repeat domain (WD7), which is involved in substrate recognition (*Skaar et al., 2013*). *De novo* missense and nonsense variants in *FBXW11* have been previously described in patients with a neurodevelopmental syndrome with abnormalities of the digits, jaw and eyes (*Holt et al., 2019*). These variants were located in WD1, WD4, and WD6 and have been shown to compromise substrate recognition or binding of the Wnt and Hedgehog signalling developmental pathways. We hypothesised a specific functional effect on NF-κB signalling in our patient with a distinct autoinflammatory phenotype. In peripheral blood mononuclear cells (PBMC) extracted from the patient, we demonstrated that the phosphorylation of the NF-κB subunit p65 was constitutively higher in monocytes and CD8+ T cells as compared to a healthy control, which suggests a functional effect of the *FBXW11* variant. This effect is further substantiated by the observation of increased p65 phosphorylation and downstream production of IL-1β after stimulation with pathogens and a TLR3 ligand in the patient. However, a note of caution should be made regarding n=1 studies, as we cannot exclude that the difference is due to normal inter-individual biological variability. These results suggest that NF-κB signalling was aberrantly increased in the patient, a mechanism that has been shown to be involved in the pathogenesis of other monogenic autoinflammatory disorders known as relopathies (*van der Made et al., 2020*). However, the exact mutational mechanism that explains the different phenotype compared to the previous cases with neurodevelopmental disease remains unclear. The most likely explanation would be a differential effect on the specific functions of the SCF complex that could be cell-type dependent rather than a difference in tissue-specific isoform expression, since the DNV affects all three high quality protein-coding transcripts that produce the most abundant isoforms (*Consortium, 2020*). Further experiments addressing the effect of this DNV on IκBα degradation, substrate recognition and TrCP protein abundance should be undertaken to provide conclusive evidence.

To our knowledge, two other cohort studies have systematically performed trio-based sequencing in IEI patients as part of their study design, although patients were not pre-selected based on sporadic phenotypes (*Stray-Pedersen et al., 2017*; *Simon et al., 2020*). Stray-Pedersen *et al.* conducted a large international cohort study to investigate the benefit of WES in IEI patients from 278 families, which included 39 patient-parent trios (*Stray-Pedersen et al., 2017*). The authors reported a molecular diagnosis in 40% of the patients, including 15 (13.6%) *de novo* mutations, of which 4 were identified by trio-based analysis and 11 after segregation analysis. The additional value of trio-based sequencing is indicated by the higher detection rate compared to that of the single cases followed by segregation analysis of candidate variants (44 vs 36%), as well as the discovery of potentially novel IEI genes or expansion of the immunological phenotype. Furthermore, Simon *et al.* performed WES in a cohort of

## Box 1.

Proposed indications for trio-based sequencing in patients with inborn errors of immunity.

1. Clinical features with a high *a priori* chance for a causative pre- or post-zygotic *de novo* variant (DNV)

   a. Sporadic and ultra-rare

   b. Early-onset (infancy/childhood)

   c. Severe symptoms, often involving organs other than the immune system

2. Clinical features with a high *a priori* chance for a causative somatic DNV acquired during life

   a. Late-onset (adolescence/adulthood)

   b. Severe symptoms, often involving signs of autoinflammation, immune dysregulation and/or bone marrow abnormalities

   c. Evidence for immune cell- or bone marrow lineage-specific dysfunction (i.e. myeloid cells [*Beck et al., 2020*], lymphoid cells [*Wolach et al., 2005*])

106 IEI patients with a consanguineous background, including 26 patient-parent trios (*Simon et al., 2020*). A molecular diagnosis was established in 70% of the patients, including 13 (17.6%) *de novo* mutations, although it is unclear whether these variants were identified through trio-based sequencing or the segregation analysis that was performed for each variant. The authors conclude that trio-based sequencing does not lead to additional diagnostic benefit, although it should be argued that this is also not expected in a cohort of predominantly consanguineous patients (62.2%) with a higher *a priori* chance of autosomal recessive (AR) disease.

Multiple studies have highlighted the potential benefits of routine trio-based sequencing in IEI patients over single WES (*Meyts et al., 2016*; *Arts et al., 2019*; *Vorsteveld et al., 2021*; *Chinn et al., 2020*). These advantages apply mostly to patients with sporadic, severe phenotypes in particular, as has been shown for other rare diseases such as neurodevelopmental disorders (*Kaplanis et al., 2020*). Trio-based sequencing constitutes an unbiased way to identify rare, coding DNVs that are by definition strong candidate variants. It could therefore improve candidate variant prioritisation both during *in silico* gene panel analysis as part of routine diagnostics, as well as for research-based exome-wide analysis. Furthermore, targeted DNV analysis could improve the detection of somatic variants, which is especially relevant in the field of monogenic autoinflammatory disorders (*van der Made et al., 2022*). Somatic variants can be successfully identified by trio-based WES (*de Koning et al., 2015*). However, this specific DNV subtype can be missed during routine analysis especially if the variant allele fraction (VAF) is below the set threshold during standard variant filtering, which is not required to filter out false-positive variants for a condensed set of potential DNVs. In this study, no candidate DNVs with a VAF below the set threshold of 20% were found in established IEI genes. Another advantage of trio-based sequencing is that it provides direct segregation of inherited variants and enables determination of autosomal recessive compound heterozygosity or X-linked recessive disease as the causative disease mechanism.

Based on the results of this study as well as evidence from other studies including those from other rare disease fields, we suggest that trio-based sequencing should be part of the routine evaluation of patients with a sporadic IEI phenotype (*Box 1*). An exome-wide analysis should be conducted to identify potentially novel disease genes in cases with a negative diagnostic WES result in whom a strong clinical suspicion for an underlying monogenic cause remains. Thus far, the relative proportion of DNVs among IEI patients with a genetic diagnosis, estimated to be around 6–14%, seems modest compared to other rare disease fields (i.e. >80% in neurodevelopmental disorders (NDDs)) (*Brunet et al., 2021*). There are several explanations for this difference that suggest that the true contribution of DNVs is higher than currently appreciated. Most importantly, much more experience has been

gained with DNV assessment in the field of NDDs. Despite a steep increase in the total diagnostic rate (*Vissers et al., 2010*; *de Ligt et al., 2012*; *Deciphering Developmental Disorders Study, 2017*) and the identification of 285 developmental disorder (DD)-associated DNVs, modelling suggests that more than 1000 DD-associated genes still remain to be discovered (*Kaplanis et al., 2020*). As more trio-based sequencing data will be generated from suspected IEI patients, the field should undertake larger-scale analyses that leverage existing statistical models from the field of NDDs/DDs, including models for gene/exon level enrichment and the identification of gain-of-function nucleotide clusters (*Kaplanis et al., 2020*). Moreover, there is still a bias towards AR disease genes in the IEI field, while this imbalance is shifting with the discovery of an increasing number of autosomal dominant (AD) disease genes (*van der Made et al., 2020*). Trio-based sequencing could accelerate the discovery of mutations in novel AD IEI genes.

Inborn errors of immunity constitute a large group of heterogeneous disorders with differences in the expected contribution of DNVs. The *a priori* probability for the identification of a DNV will be highest in patients with early-onset, severe phenotypes, such as the combined immunodeficiencies (CID), especially CIDs with syndromic features, and patients with autoinflammatory syndromes and/or immune dysregulation with autoimmunity (*Box 1*). Although most of the reported genes underlying CIDs follow AR inheritance patterns, many genes following AD and X-linked (dominant) inheritance patterns have been described in recent years (*Tangye et al., 2020*). The genes affected in these disorders possess high intolerance for loss-of-function mutations and essential biological functions. As expected, the DNVs in this category reported to date act through mechanisms of haploinsufficiency (i.e. *RELA*, pLI 0.999), dominant-negative interference (i.e. *IKZF1*, pLI 0.999 *Kuehn et al., 2016*; *STAT3*, pLI 1.000 *Holland et al., 2007*) or complete deficiency in hemizygotic males (i.e. *WAS*, pLI 0.999 *Howard et al., 2016*; *IL2RG*, pLI 0.992 *Moya-Quiles et al., 2014*). Some heterozygous DNVs can also cause CID through hypermorphic effects at protein level (i.e. *RAC2*, pLI 0.966 *Hsu et al., 2019*). Trio-based sequencing should also be considered in patients with sporadic autoinflammatory syndromes and/or autoimmunity, even when presenting at an adult age that could suggest somatic *de novo* mutations. In these patients, various pathogenic DNVs in different genes have already been described, originating both from the germline (*PLCG2, STAT1*) and soma (i.e. *NLRP3, UBA1, TLR8*) (*Liu et al., 2011*; *Mensa-Vilaro et al., 2016*; *Aluri et al., 2021*; *Beck et al., 2020*; *van der Made et al., 2022*; *Zhou et al., 2012*). These genes do not necessarily have high constraint for LoF mutations, but they possess nucleotide clusters that are highly conserved and intolerant to variation, encoding protein domains with important regulatory functions.

This explorative study has a number of limitations. First, the sample size precludes a reliable estimation of the prevalence of DNVs among patients with sporadic IEIs. Furthermore, the strict diagnostic rate of both inherited variants and (likely) pathogenic DNVs in our cohort is limited compared to other studies. It has been previously reported that the diagnostic yield of WES for IEI patients varies widely from 10 to 79% (*Vorsteveld et al., 2021*). This study reports (likely) pathogenic variants in 22 cases (17.9%), of which 10 (8.1%) received a definitive molecular diagnosis for their immunological phenotype. In addition to inherent technical shortcomings of WES, including uneven coverage of coding regions and GC bias and also the inability to explore the non-coding space (*Meyts et al., 2016*), the most likely explanation for a relatively low diagnostic yield in our study is the patient selection and the primary focus on DNVs, which constitute only a fraction of disease-causing variants. We excluded patients with suspected inherited disease but chose not to apply any other selection criteria in order to study a representative cross-section of suspected IEI patients in our centre in whom WES was performed. As a result, patients were included even if the *a priori* chance of an IEI was limited but to be ruled out in the differential diagnosis (i.e. new-born screening shows low T cell receptor excision circles (TRECs)). Moreover, compared to other cohorts, the percentage of patients with syndromal CIDs, autoinflammatory syndromes and immune dysregulation was relatively high and could influence the generalisability of our results. Lastly, the functional effect of most candidate DNVs were not evaluated. As DNVs have a high chance of being deleterious, functional experiments should always be attempted to validate the predicted effect. The candidate DNVs in potentially novel IEI genes were shared on GeneMatcher in order to find similar cases that could motivate further investigation into the underlying mechanisms (*Acuna-Hidalgo et al., 2016*; *Sobreira et al., 2015*).

In conclusion, we applied trio-based WES in a retrospective cohort of 123 patients with suspected, sporadic IEI, leading to the identification of 14 DNVs with a possible or likely chance of pathogenicity.

Amongst the candidate DNVs in potentially novel IEI genes, additional functional evidence was provided in support of a pathogenic role for the DNV in *FBXW11* in a patient with an autoinflammatory phenotype. We advocate the structural implementation of trio-based sequencing in the diagnostic evaluation of patients with sporadic IEI. With decreasing costs of exome sequencing, this approach could improve the diagnostic rate of IEI and advance IEI gene discovery.

## Acknowledgements

We thank the Bioinformatics group of the Genome Diagnostics division of the department of Human Genetics and the Radboud Genomics Technology Center of the Radboud University Medical Center for the sharing, annotation and pseudonymisation of whole exome sequencing datasets of patients and their parents included in this study. Furthermore, we acknowledge all members of the multidisciplinary immunogenetics sign-out meeting of the University Medical Centers from Nijmegen and Maastricht. The authors also acknowledge support from several funding parties. MG Netea was supported by an ERC Advanced Grant (No. 833247) and a Spinoza Grant of the Netherlands Organization for Scientific Support. This research was also part of a Radboud Institute for Molecular Life Sciences PhD grant (to M G Netea). F L van de Veerdonk was supported by a ZonMW Vidi grant and HDM-FUN EU H2020. A Hoischen was supported by the Solve-RD project of the European Union's Horizon 2020 research and innovation programme (No. 779257). Funding This research was supported by grants from the European Union, ZonMW and the Radboud Institute for Molecular Life Sciences.

## Additional information

### Competing interests

Frank L van de Veerdonk: Reviewing editor, *eLife*. The other authors declare that no competing interests exist.

### Funding

| Funder | Grant reference number | Author |
| --- | --- | --- |
| European Research Council | No. 833247 | Mihai G Netea |
| ZonMw | Spinoza Grant | Mihai G Netea |
| Radboud Institute for Molecular Life Sciences | Internal grant | Mihai G Netea |
| ZonMw | Vidi | Frank L van de Veerdonk |
| H2020 European Research Council | HDM-FUN | Frank L van de Veerdonk |
| H2020 European Research Council | Solve-RD (No. 779257) | Alexander Hoischen |

The funders had no role in study design, data collection and interpretation, or the decision to submit the work for publication.

### Author contributions

Anne Hebert, Data curation, Formal analysis, Investigation, Visualization, Methodology, Writing – original draft, Writing – review and editing; Annet Simons, Data curation, Formal analysis, Investigation, Methodology, Writing – review and editing; Janneke HM Schuurs-Hoeijmakers, Evelien Zonneveld-Huijssoon, Stefanie SV Henriet, Ellen JH Schatorjé, Esther PAH Hoppenreijs, Erika KSM Leenders, Etienne JM Janssen, Gijs WE Santen, Sonja A de Munnik, Resources, Writing – review and editing; Hans JPM Koenen, Resources, Formal analysis, Visualization, Methodology, Writing – review and editing; Simon V van Reijmersdal, Investigation, Methodology, Writing – review and editing; Esther van Rijssen, Resources, Investigation, Methodology, Writing – review and editing; Simone Kersten, Resources, Investigation, Writing – review and editing; Mihai G Netea, Supervision, Funding acquisition, Methodology, Writing – review and editing; Ruben L Smeets, Resources, Formal analysis,

Methodology, Writing – review and editing; Frank L van de Veerdonk, Supervision, Funding acquisition, Writing – review and editing; Alexander Hoischen, Conceptualization, Formal analysis, Supervision, Funding acquisition, Investigation, Writing – original draft, Project administration, Writing – review and editing; Caspar I van der Made, Conceptualization, Data curation, Formal analysis, Supervision, Investigation, Visualization, Methodology, Writing – original draft, Project administration, Writing – review and editing

## Author ORCIDs
Anne Hebert http://orcid.org/0000-0002-8945-015X
Simone Kersten http://orcid.org/0000-0002-0251-5564
Mihai G Netea http://orcid.org/0000-0003-2421-6052
Frank L van de Veerdonk http://orcid.org/0000-0002-1121-4894
Alexander Hoischen http://orcid.org/0000-0002-8072-4476
Caspar I van der Made http://orcid.org/0000-0003-0763-4017

## Ethics
Human subjects: Patients and their parents provided written informed consent for in silico inborn errors of immunity whole exome sequencing gene panel analysis with or without exome-wide variant analysis in line with the diagnostic procedure and clinical question, as approved by the Medical Ethics Review Committee Arnhem-Nijmegen (2011/188 and 2020-7142). This research is in compliance with the principles of the Declaration of Helsinki.

## Decision letter and Author response
Decision letter https://doi.org/10.7554/eLife.78469.sa1
Author response https://doi.org/10.7554/eLife.78469.sa2

---

# Additional files

## Supplementary files
• Transparent reporting form

## Data availability
The code used to filter DNA sequencing data for candidate *de novo* mutations (DNMs) and to generate output files is provided in Figure 1—source code 1. Source data linked to Figure 1—figure supplement 1 is provided as an additional, numerical data file. Source data for candidate DNM evaluation is provided in Figure 1—source data 2. Source data linked to Figure 2—figure supplement 1A is an uncropped, raw gel image used to create this figure. Source data linked to Figure 2B–D is provided as an additional, numerical data file. Raw DNA sequencing data of patients are not publicly available as it is confidential human subject data that would compromise anonymity. Researchers that are interested to access the sequencing data of our cohort are advised to contact the corresponding author, A Hoischen (alexander.hoischen@radboudumc.nl). Anonymized subject data will be shared on request from qualified investigators for the purposes of replicating procedures and results, and for other non-commercial research purposes within the limits of participants' consent. Any data sharing will also require evaluation of the request by the regional Arnhem and Nijmegen Ethics Committee and the signature of a data transfer agreement (DTA).

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
