## [Editor Report]

This is an important paper that reports on the diagnostic utility of TRIO-based whole-exome sequencing (WES) for patients with suspected monogenic inborn errors of immunity, which is supported by solid data. This manuscript will be of particular interest to medical geneticists, immunologists, and physicians working with patients with primary immunodeficiencies.

---

## [Decision Letter]

**Decision letter after peer review:**

Thank you for submitting your article "Trio-based whole exome sequencing in patients with suspected sporadic inborn errors of immunity: a retrospective cohort study" for consideration by *eLife*. Your article has been reviewed by 2 peer reviewers, and the evaluation has been overseen by a Reviewing Editor and Mone Zaidi as the Senior Editor. The reviewers have opted to remain anonymous.

Essential revisions:

1) As commented by Reviewer 1, please check the filtering strategy for DNV calls to filter false positives.

2) The novel identified candidate variants should be confirmed by Sanger sequencing or some other methods/tools (CNVs). This will give a sense of any false variant calls in this type of analysis.

*Reviewer #1 (Recommendations for the authors):*

Genomics analysis of rare diseases, including IEI, is valuable to the field despite the limited sample size. However, the current manuscript seems to focus on describing affected genes (though a functional experiment on one candidate). I think the paper will be more suitable for readers in clinical immunology or medical genetics.

*Reviewer #2 (Recommendations for the authors):*

1. Page 5, line 28, "<20% variant allele frequency". Do the authors mean Variant Allele Fraction? The filtering pipeline excluded somatic variation below 20% VAF, which is understandable as decreasing the VAF cutoff introduces a lot of noise; nevertheless, in some diseases, somatic variations are found at 5-20% VAF. This could be commented on in the Discussion.

2. Page 11, line 13: the wording in this sentence is too strong. Finding a heterozygous pathogenic or a likely pathogenic variant in recessively inherited disease-associated genes does not constitute " a likely molecular diagnosis." Unless there is evidence that at least some of these pathogenic and LP variants affect the same pathway and have a synergistic effect.

3. What is the percentage of patients with a confirmed molecular diagnosis? I can count only 6 patients in the table 3 (10,12, 59,91,105,122). Please correct me if I am wrong.

4. Finding small CNV's (for example 1-2 exon deletions) is hard due to a high false-positive rate. If only one tool was used (i.e., CoNIFER), did the authors use any additional method to confirm these de novo CNVs in patients 21, 69, and 115? It is unclear to me whether any of these CNVs have been previously identified/reported in some databases.

5. During the variant filtering step, only de novo nonsynonymous coding and splice region variants were kept. The authors mentioned analyzing splice-site DNVs specifically using the Alamut software, which provides a few prediction tools. The authors may consider using other algorithms, such as SpliceAI, which appears to be highly reliable for prioritizing variants. (PMID: 30661751)

6. Table 4, I question the pathogenicity predictions regarding the KMT2C frameshift variant. The CADD score is only 14.7, and according to UCSC as well as polyphen-2, this residue is not conserved between species (Serine to Histidine between human and mouse).

7. Page 17, line 9: although most proteasome-associate diseases are recessively inherited, there are examples of a dominantly inherited mutation in PSMB9. (PMID 33727065 and 34819510)

8. Page 21, line 13: are there different isoforms of FBXW11? Cell-specific expression of various isoforms can potentially explain how LoF variants in this gene can be associated with a neurodevelopmental phenotype in some patients and an immunological disease in other patients. (e.g., CDC42-associated diseases, UBA1-associated diseases).

9. Page 24, line 10: there is a typo in this sentence " soma", it should be "somatic".

10. The authors may discuss other possible explanations for the lower-than-expected diagnostic rate in this cohort and similar cohorts of patients with suspected monogenic immunological diseases. Possible explanations are uneven coverage of coding regions (especially exon 1 tends to be poorly covered), the fact that only protein-coding regions are captured by WES, somatic mutations in the early post-zygotic stage (PMID: 32750333), or some other explanations. Reduced penetrance is another factor, which is not pertinent to this study, as this study focuses on the identification of de novo variants. Expectedly, the diagnostic rate will be higher when working with patients with recessively inherited PID and from founder populations and this may explain the diagnostic rate of 75% in some studies.

11. Some sentences are repetitive in the Result section and Discussion e.g., the discussion on the PSMB10, RELA, and NLRP3 variants.

---

## [Author Response]

Essential revisions:1) As commented by Reviewer 1, please check the filtering strategy for DNV calls to filter false positives.

We agree with Reviewer 1 that filtering for rare variants, even more so for DNVs, is prone to falsepositive findings. Our filtering strategy is in accordance with other DNV filtering strategies in literature [1-3]. While this filtering strategy minimizes the likelihood for false-positive findings, it ensures that only a fraction of true positives are missed. A very low variant allele frequency should be permitted, especially for variants in genes known to cause inborn errors of immunity. DNVs in these genes that are found in presumed healthy individuals in population databases might still be pathogenic due to decreased penetrance, adult-onset development of the disease or postzygotic occurrence. Apart from standard filters on variant allele frequency and quality, we have applied several approaches to minimize the risk of false-positive DNVs:

To minimize the contribution of local genetic variation and sequencing artefacts to potential false-positive variant calls, we have filtered variants both on the Dutch population database (GoNL) and our in-house database (containing >60,000 exomes) variant allele frequencies.Additionally, the automated DNV calling is based on >2% alternate allele reads in either parent, meaning that the DNVs are high confidence calls.Variants were further filtered based on an annotated function in the immune system, which increased the likelihood of finding true-positive DNVs.The final 14 candidate DNVs and variants reported by the diagnostic department were inspected visually with the Integrative Genomics Viewer (IGV) and/or Alamut Visual software to exclude false-positive variants. We have added Author response image 1–Author response image 17 , our final 14 candidate DNVs, for the inspection by the Editors and reviewers, and could add this information to the manuscript supplement if judged to be helpful.

**Author response image 1. sa2fig1:** Patient 1: PSMB10 Chr16(GRCh37):g.67968809C>T NM_002801.3:c.601G>A p.(Gly201Arg). Score <500, but validated by an independent test (Sanger sequencing).

**Author response image 2. sa2fig2:** Patient 9: RPL27A Chr11(GRCh37):g.8707228T>C NM_000990.4:c.322T>C p.(Tyr108His). Validated by an independent test (identified by single and trio-based WES).

**Author response image 3. sa2fig3:** Patient 21: seq[GRCh37] del(17)(q25.3qter) NC_000017.10:g.(80544076_80544938)_qterdel. Validated by independent test (identified by two different enrichment kits: Agilent Technologies and Twist Bioscience).

**Author response image 4. sa2fig4:** Patient 27: TAOK2 Chr16(GRCh37):g.29997683C>T NM_016151.3:c.2090C>T p.(Ala697Val).

**Author response image 5. sa2fig5:** Patient 28: KCTD9 Chr8(GRCh37):g.25292997C>T NM_017634.3:c.695G>A p.(Arg232His). Score <500, but validated by an independent test (Sanger sequencing).

**Author response image 6. sa2fig6:** Patient 49: DDX1 Chr2(GRCh37):g.15769802dup NM_004939.2:c.1952dup p.(Trp652fs).

**Author response image 7. sa2fig7:** Patient 52: SCRIB Chr8(GRCh37):g.144874432C>T NM_182706.4:c.4472G>A p.(Arg1491Gln). Score <500 and not validated by an independent test.

**Author response image 8. sa2fig8:** Patient 53: FBXW11 Chr5(GRCh37):g.171295802T>C NM_012300.2:c.1468-2A>G p.?.

**Author response image 9. sa2fig9:** Patient 58: CTCF Chr16(GRCh37):g.67645905G>T NM_006565.4:c.833G>T p.(Arg278Leu).

**Author response image 10. sa2fig10:** Patient 59: NLRP3 Chr1(GRCh37):g.247587794C>T NM_001079821.2:c.1049C>T p.(Thr350Met). Score <500, but validated by an independent test (identified prior to trio-based WES in a different hospital).

**Author response image 11. sa2fig11:** Patient 59: NLRP3 Chr1(GRCh37):g.Patient 69: seq[GRCh37] dup(22)(q11.21q11.21) NC_000022.10:g.(18775421_18893960)_(21414845_21576183)dup.

**Author response image 12. sa2fig12:** Patient 75: FUBP1 Chr1(GRCh37):g.78435621A>C NM_001303433.1:c.199T>G p.(Leu67Val). Validated by an independent test (identified by single and trio-based WES).

**Author response image 13. sa2fig13:** Patient 78: KMT2C Chr7(GRCh37):g.151860074del NM_170606.2:c.10588del p.(Ser3530Leufs*3).

**Author response image 14. sa2fig14:** Patient 108: NSD2 Chr4(GRCh37):g.1959681_1959687delinsTTTTTCT NM_133330.2:c.2903_2909delinsTTTTTCT p.(Arg968_Arg970delinsLeuPheLeu). Validated by independent test (WES on different tissue (skin biopsy)).

**Author response image 15. sa2fig15:** Patient 115: seq[GRCh37] dup(X)(q13).

**Author response image 16. sa2fig16:** Patient 118: RUNX3 Chr1(GRCh37):g.25256227C>T NM_004350.2:c.133G>A p.(Gly45Arg).

**Author response image 17. sa2fig17:** Patient 119: RELA Chr11(GRCh37):g.65423234C>T NM_021975.3:c.959-1G>A p.?.

Integrative Genomics Viewer (IGV) and/or Alamut Visual software screenshots of the 14 final candidate de novo variants and 3 de novo copy number variants

In conclusion, we are convinced that the used DNV filtering strategy already minimizes the chance of false-positives; and average number of DNVs are in line with recent literature (Figure 1 – source data 3; [4]). Nevertheless, we have adjusted the filtering strategy according to the reviewer’s suggestion by increasing the variation reads from 5 to 10, as this stricter cut-off is also mentioned by some studies in literature [3, 5, 6]. This reduced the total amount of rare DNVs to 172 (previously 187). In addition, we have clarified in the text that our analysis includes 24 variants that were present at very low allele frequencies in population databases (page 13 line 4). We have updated the text (page 5 line 7-8 and 30, page 6 lines 6-8, page 13 lines 3-5 and 8, page 15 line 1, page 23 line 12), and Figure 1 with corresponding table supplement 2 and 3 as well as figure supplement 1.

2) The novel identified candidate variants should be confirmed by Sanger sequencing or some other methods/tools (CNVs). This will give a sense of any false variant calls in this type of analysis.

We agree that for this type of analysis it is very important to keep the possibility of false-positive variant calls as limited as possible. However, the use of Sanger sequencing to confirm SNV candidate variants is no longer standard of practice – in our nor other laboratories as large validation studies preceded the current effort. SNVs called in next-generation sequencing techniques such as whole exome sequencing have been shown to reach a concordance of 99.965% with Sanger sequencing, indicating a very limited chance on false-positives [7]. To further support the absence of any type of bias that would give rise to false-positive DNVs (low quality base calls, read end artifacts, strand bias artifacts, erroneous alignment to low complexity regions, and paralogous alignment of reads) [8], we have added IGV and/or Alamut screenshots of the 14 final 14 DNVs, remaining as the most promising variants, at the end of this rebuttal. Furthermore, our diagnostics department follows standard practice guidelines (ISO15189) that dictate the performance of validations using other techniques if quality scores are <500, as the chances of a false-positive variant call is very low if this score exceeds 500. According to these guidelines, the DNVs identified in patient 1, 28, 52 and 59 would require validation for diagnostic reporting. For three of these patients (1, 28 and 59) as well as patients 9, 75 and 108 independent tests such as re-sequencing, sequencing of other tissue samples or Sanger sequencing were performed and confirmed the presence of the identified DNVs. All other DNVs fulfil diagnostic quality criteria and would be shared in diagnostic reports without any further validation.

Finally, the de novo copy number variants (CNVs) of patient 21, 69 and 115, were also reported as high confident calls by the diagnostic division of our department according to standard operating procedures, please refer to the attached IGV and/or Alamut screenshots. The high quality scores were most importantly based on the large CNV sizes: patient 21 del17q 500kb, patient 69 dup22q 2.5MB and patient 115 dupXq 400-500kb. Moreover, the CNV in patient 21 was identified by two different enrichment kits, increasing the likelihood that this CNV is a true-positive. Lastly, the CNV in patient 69 is involved in a recurrent microduplication syndrome (Chromosome 22q11.2 micro-duplication syndrome, OMIM #608363) with well-established breakpoints in low-copy repeat elements mediated by nonallelic homologous recombination, which makes true nature of this CNV very likely.

Reviewer #1 (Recommendations for the authors):Genomics analysis of rare diseases, including IEI, is valuable to the field despite the limited sample size. However, the current manuscript seems to focus on describing affected genes (though a functional experiment on one candidate). I think the paper will be more suitable for readers in clinical immunology or medical genetics.

We thank the reviewer for the overall positive assessment of our work. We leave it up to the expertise of the Editor to choose the most appropriate research area within *eLife* that would reach this readership.

Reviewer #2 (Recommendations for the authors):1. Page 5, line 28, "<20% variant allele frequency". Do the authors mean Variant Allele Fraction? The filtering pipeline excluded somatic variation below 20% VAF, which is understandable as decreasing the VAF cutoff introduces a lot of noise; nevertheless, in some diseases, somatic variations are found at 5-20% VAF. This could be commented on in the Discussion.

We indeed meant to express “variant allele fraction”, which has now been corrected in the revised manuscript. Moreover, the reviewer is right in pointing out that somatic variants could be missed with a VAF cut-off of 20%. However, we have checked that no filtered DNVs with a VAF below 20% were missed in established inborn errors of immunity genes. We have added a sentence specifying this in the Materials and methods (page 5 line 31 – page 6 line 1) and Discussion section (page 24 lines 20-21).

2. Page 11, line 13: the wording in this sentence is too strong. Finding a heterozygous pathogenic or a likely pathogenic variant in recessively inherited disease-associated genes does not constitute " a likely molecular diagnosis." Unless there is evidence that at least some of these pathogenic and LP variants affect the same pathway and have a synergistic effect.

We think this might be based on a misunderstanding, as we do not suggest that the finding of a heterozygous pathogenic or likely pathogenic variant in a recessive disease gene would constitute a diagnosis, but merely carriership. We have updated Table 2 to make it more comprehensible.

3. What is the percentage of patients with a confirmed molecular diagnosis? I can count only 6 patients in the table 3 (10,12, 59,91,105,122). Please correct me if I am wrong.

This too might be based on a misunderstanding. We have rephrased the corresponding sentence in the revised manuscript to increase comprehensibility (page 12 lines 19-21) and we have additionally updated Table 2.

4. Finding small CNV's (for example 1-2 exon deletions) is hard due to a high false-positive rate. If only one tool was used (i.e., CoNIFER), did the authors use any additional method to confirm these de novo CNVs in patients 21, 69, and 115? It is unclear to me whether any of these CNVs have been previously identified/reported in some databases.

As elaborated in our response on page 2, recommendation 2, the de novo CNVs in patients 21, 69 and 115 were all based on high-confidence calls. Due to the high quality scores, the CNVs were not validated by additional sequencing approaches. The CNVs were also manually inspected, please refer to the attached IGV and/or Alamut screenshots. Moreover, the CNV in patient 69 had been previously reported as a cause for 22q11.2 microduplication syndrome (page 12 lines 8-10, Table 2 – source data 1). The CNV identified in patient 21 was encountered in DECIPHER, a reference to which has been added to the revised manuscript (page 12 lines 7-8, Table 2 – source data 1). The CNV of patient 115 was not identified in any database (page 12 lines 10-12). However, it was previously described in another case of our hospital, although the phenotypic overlap was unclear.

5. During the variant filtering step, only de novo nonsynonymous coding and splice region variants were kept. The authors mentioned analyzing splice-site DNVs specifically using the Alamut software, which provides a few prediction tools. The authors may consider using other algorithms, such as SpliceAI, which appears to be highly reliable for prioritizing variants. (PMID: 30661751)

We thank the reviewer for this useful comment. We have added the SpliceAI scores to Table 3 and now mention its use in the Materials and methods section (page 6 line 9).

6. Table 4, I question the pathogenicity predictions regarding the KMT2C frameshift variant. The CADD score is only 14.7, and according to UCSC as well as polyphen-2, this residue is not conserved between species (Serine to Histidine between human and mouse).

Since variant-specific metrics are not useful to assess the functional impact of predicted loss-offunction variants, we have removed these metrics from Table 4 for the predicted loss-of-function variants. In contrast, gene-specific metrics are much more informative. As stated in Table 4, the LOEUF score of KMT2C is 0.12, which indicates a strong constraint against loss-of-function mutations.

7. Page 17, line 9: although most proteasome-associate diseases are recessively inherited, there are examples of a dominantly inherited mutation in PSMB9. (PMID 33727065 and 34819510)

We thank the reviewer and have adapted the suggestion by adding a section on the dominantly inherited *PSMB9* mutation: “PRAAS is … in mice.” (page 22 lines 10-16).

8. Page 21, line 13: are there different isoforms of FBXW11? Cell-specific expression of various isoforms can potentially explain how LoF variants in this gene can be associated with a neurodevelopmental phenotype in some patients and an immunological disease in other patients. (e.g., CDC42-associated diseases, UBA1-associated diseases).

FBXW11 indeed has multiple isoforms, as can be observed in Figure 2 —figure supplement 1. It has three high quality protein-coding transcripts that produce the most abundant isoforms, and all are affected by the splice-acceptor DNV. The cell-type or tissue-specific expression does not seem to differ greatly among the isoforms (GTEx Portal, v8), but it could be that the specific functions of the SCF complex in which FBXW11 participates are cell-type specific. We have now also stated this in the discussion: “However, the exact mutational mechanism … most abundant isoforms.” (page 21 lines 19-24).

9. Page 24, line 10: there is a typo in this sentence " soma", it should be "somatic".

We have not corrected this as we actually meant to express the noun, which is ‘soma’.

10. The authors may discuss other possible explanations for the lower-than-expected diagnostic rate in this cohort and similar cohorts of patients with suspected monogenic immunological diseases. Possible explanations are uneven coverage of coding regions (especially exon 1 tends to be poorly covered), the fact that only protein-coding regions are captured by WES, somatic mutations in the early post-zygotic stage (PMID: 32750333), or some other explanations. Reduced penetrance is another factor, which is not pertinent to this study, as this study focuses on the identification of de novo variants. Expectedly, the diagnostic rate will be higher when working with patients with recessively inherited PID and from founder populations and this may explain the diagnostic rate of 75% in some studies.

We would like to thank Reviewer 2 for these suggestions. We have integrated some of these ideas into the Discussion section (page 26 lines 12-15).

11. Some sentences are repetitive in the Result section and Discussion e.g., the discussion on the PSMB10, RELA, and NLRP3 variants.

We have addressed this by reducing the respective sections in the Results section and by re-writing parts of the paragraphs in the Discussion section. Please refer to the revised manuscript.

References

1. Olfson, E., et al., Whole-exome DNA sequencing in childhood anxiety disorders identifies rare de novo damaging coding variants. Depress Anxiety, 2022. 39(6): p. 474-484.

2. Pedersen, B.S., et al., Effective variant filtering and expected candidate variant yield in studies of rare human disease. NPJ Genom Med, 2021. 6(1): p. 60.

3. Oud, M.S., et al., *A* de novo *paradigm for male infertility.* Nat Commun, 2022. 13(1): p. 154.

4. Kaplanis, J., et al., Evidence for 28 genetic disorders discovered by combining healthcare and research data. Nature, 2020. 586(7831): p. 757-762.

5. Dong, W., et al., Exome Sequencing Implicates Impaired GABA Signaling and Neuronal Ion Transport in Trigeminal Neuralgia. iScience, 2020. 23(10): p. 101552.

6. Homsy, J., et al., de novo mutations in congenital heart disease with neurodevelopmental and other congenital anomalies. Science, 2015. 350(6265): p. 1262-6.

7. Beck, T.F., et al., Systematic Evaluation of Sanger Validation of Next-Generation Sequencing Variants. Clin Chem, 2016. 62(4): p. 647-54.

8. Koboldt, D.C., Best practices for variant calling in clinical sequencing. Genome Med, 2020. 12(1): p. 91.

9. Wang, T., et al., Integrated gene analyses of de novo mutations from 46,612 trios with autism and developmental disorders. BioRxiv, 2021.

10. Hoischen, A., N. Krumm, and E.E. Eichler, Prioritization of neurodevelopmental disease genes by discovery of new mutations. Nat Neurosci, 2014. 17(6): p. 764-72.